# Measuring the Whole Wall Thickness of the Common Femoral Vein as a Distinctive Diagnostic Tool to Distinguish Behçet’s Disease Presenting with Oral Ulcers from Recurrent Aphthous Stomatitis

**DOI:** 10.3390/diagnostics13162705

**Published:** 2023-08-19

**Authors:** Adem Erturk, Yasin Sarıkaya, Hüseyin Coşkun, Çağrı Turan

**Affiliations:** 1Division of Rheumatology, Department of Internal Medicine, Faculty of Medicine, Afyonkarahisar Health Sciences University, Afyonkarahisar 03030, Turkey; 2Department of Radiology, Faculty of Medicine, Afyonkarahisar Health Sciences University, Afyonkarahisar 03030, Turkey; yasin.sarikaya@afsu.edu.tr (Y.S.); huseyin.coskun@afsu.edu.tr (H.C.); 3Department of Dermatology and Venereology, Faculty of Medicine, Afyonkarahisar Health Sciences University, Afyonkarahisar 03030, Turkey; cagri.turan@afsu.edu.tr

**Keywords:** Behçet’s disease, oral ulcers, recurrent aphthous stomatitis, ultrasonography, differential diagnosis, vein wall thickness, vein whole-wall thickness, vein intima–media thickness, diagnostic

## Abstract

Recurrent oral ulcers, which are the first and most common manifestation of Behçet’s disease (BD), have several etiological causes but are often idiopathic and known as recurrent aphthous stomatitis (RAS). In recent years, publications have drawn attention to the fact that whole-wall thickness (WWT) and intima–media thickness (IMT) measurements of the common femoral vein (CFV) may be useful in the diagnosis of BD, which are independent of organ involvement. In this study, the usefulness of the WWT and IMT of the CFV measured by venous Doppler ultrasound (US) was investigated in differentiating BD presenting with oral ulcers from RAS. Patients with BD (*n* = 84), patients with RAS (*n* = 85), and healthy controls (HCs) (*n* = 70) were the three groups included. Both the right and left WWT of the CFV were significantly higher in the BD compared with the HCs and RAS groups (*p* < 0.001). The diagnostic cutoff values of ≥0.58 mm for the WWT of the CFV performed well against both the patients with RAS and HCs for the discrimination of BD (sensitivity = 79.8%, specificity = 64.7%). In patients with recurrent oral ulcers, the WWT of the CFV measurement may be a distinctive new diagnostic tool for the differential diagnosis of BD and RAS.

## 1. Introduction

Behçet’s disease (BD) is a systemic inflammatory condition of unknown etiology. It is most commonly seen in Turkey, with a prevalence of 420 patients in 100,000 individuals [1]. Although it is considered to be an orphan disease in the rest of the world, there is evidence that its prevalence may be increasing in Europe [2]. The disease usually appears in the third decade of life. Men and women are affected almost equally, although this rate may vary by country and clinic where the study was conducted [3]. In men, the disease progresses more severely, and major organ involvement is more common. BD can accelerate mortality in young men, as well as cause severe neurological damage such as vision loss, paralysis, or severe cognitive impairment. After the onset of the disease, organ involvement progresses in the form of attacks and remissions [3,4]. While attacks are more frequent and severe in the initial years of BD, they become more benign as time passes [4].

Recurrent oral ulcers (ROUs) are noted in 95–97% of patients with BD, and usually, they are the first manifestation of the disease. These ulcers occur at an average of 6–7 years before the diagnostic criteria are met [5]. Apart from ROUs, genital ulcers and uveitis are the most common clinical findings in patients with BD and are considered synonymous with the disease [2,4]. This is so much so the case that it has become a habit in our outpatient clinic that all patients presenting with recurrent oral aphthae are first evaluated for BD. However, although there are many causes other than BD, such as trauma, genetic, nutritional, infectious, immune, drug-mediated, rheumatological, and endocrinological, that can be listed in the etiology of ROUs, it is often idiopathic and known as recurrent aphthous stomatitis (RAS). RAS is the most common oral cavity disease, wherein it accounts for 25% of oral ulcers in adults and for 40% of oral ulcers in children [6]. Therefore, clinical and laboratory markers are needed to predict whether patients with only ROUs will progress to BD. In fact, the early detection of oral ulcers caused by BD will contribute to the early initiation of treatment and follow-up, as well as decrease the morbidity and mortality of BD [7]. Unfortunately, despite numerous studies on this subject, there are still no easily accessible, reliable, or generally accepted diagnostic tests that would allow us to categorize patients who are only presenting with ROUs as BD or RAS [7].

In the revised Chapel Hill Consensus Conference, BD was defined as a variable vessel vasculitis due to the involvement of both the arterial and venous vessels of all sizes [8]. The vascular involvement in BD is observed in approximately one-third of patients, and the primary involvement is the venous system [9]. Vascular manifestations are primarily the result of an impaired inflammatory response, and BD has been characterized as a natural model of thromboinflammation. Endothelial dysfunction, platelet activation, and thrombogenesis are directly sustained by neutrophil hyperactivation and neutrophil-mediated mechanisms of damage in BD [10].

Clinical assessments and imaging methods, including the Doppler ultrasound (US), computed tomography, and magnetic resonance angiography, are used to diagnose vascular abnormalities. Due to the fact that previous studies mostly focused on arterial wall thickness, there is little information available on the measurement of the venous wall inflammation in BD [11]. The first controlled Doppler US study by Alibaz-Oner et al. showed that vein wall thickness (VWT) increased in the lower extremity veins of male patients with BD, which was independent of vascular involvement [12]. Furthermore, other US studies involving different disease groups supported that the VWT increased in cases of BD [11,13,14,15,16,17]. Based on these findings, it has been suggested that the measurement of the common femoral vein (CFV) thickness can be a valuable, practical, and cost-effective diagnostic tool for diagnosing BD [9]. Furthermore, it has been reported that the thickening of the intima–media of the CFV may occur in patients with BD in addition to increased VWT [17,18].

Although oral ulcers caused by RAS and BD are morphologically indistinguishable, it is plausible that measurements of the whole-wall thickness (WWT) and intima–media thickness (IMT) of the CFV could be used to distinguish these two diseases because of possible differences in their pathological processes. In this study, we aimed to investigate the differences between the WWT and IMT of the CFV measured using Doppler US in patients with BD and RAS, as well as investigate the performance of this method in differentiating these two diseases.

## 2. Materials and Methods

### 2.1. Study Design, Setting, and Ethics

Ethical approval of this single-center, observational, and cross-sectional study was acquired from the Ethics Committee of Afyonkarahisar Health Sciences University, Afyonkarahisar, Turkey (Decision date: 5 May 2023, decision no: 2023/246). The study was performed according to the Declaration of Helsinki 1975, as revised in 2013. All data, including sociodemographic and clinical data, comorbidity status, smoking habits, disease duration, medications, and laboratory results on BD and RAS patients followed in Afyonkarahisar Health Sciences University Rheumatology and Dermatology Departments between February 2021 and January 2023, were obtained from hospital computer records and patient files (Figure 1 and Figure 2). Patients who met the study criteria were invited to the study. Patients who wanted to participate in the study were re-evaluated in terms of eligibility criteria. Doppler US was performed on 102 BD and 92 RAS patients on the day they applied to the outpatient clinic. Patients with evidence of venous insufficiency following an ultrasound and inconsistency (>0.1 mm) between the bilateral whole-wall and intima–media thickness measurements were excluded. The healthy controls were patients with hair loss in the dermatology outpatient clinic who were invited consecutively to the study. Doppler measurements were taken free of charge from the patients who gave consent. Informed consent was obtained from all participants who agreed to undergo ultrasonographic evaluations.

### 2.2. Participants and Eligibility Criteria

The study population consisted of the following three groups of patients who were admitted to the Rheumatology, Dermatology, and Venereology outpatient clinics: BD patients presenting oral ulcers (*n* = 84, BD group), idiopathic RAS patients (*n* = 85, RAS group), and healthy controls (*n* = 70, HC group).

The diagnosis of BD was made according to the International Study Group for Behçet’s Disease Criteria [19]. Patients diagnosed with BD with mucocutaneous findings but without major organ involvement such as ocular, vascular, neurological, and gastrointestinal involvement as well as patients without mucocutaneous involvement except for recurrent oral ulcers for the last three months, were included in the BD group.

The RAS group consisted of patients who met the recurrent oral ulcers diagnostic criteria recommended by Natah et al. [20], with an oral ulcer frequency of 3 or more and 8 or less per year, having recurrent oral aphthae for three or more years, and those with negative pathergy test. Patients not having a pathergy test, those with a family history of BD, inflammatory bowel disease, or celiac disease, a history of major or herpetiform aphthae, those with known HLA-B51 positivity, those with a diagnosis of incomplete BD, or those who were likely to progress to BD were not included in the RAS group.

The HC group consisted of individuals with complaints of hair loss and negative pull tests without known systemic disease. The pull test was conducted as follows: About 40 strands of hair from different areas of the scalp were grasped and gently pulled. Obtaining six or less hair was interpreted as a negative pull test (indicating normal shedding), whereas obtaining more than six hair was interpreted as a positive pull test (active hair loss) [7]. Those diagnosed with cicatricial alopecia, tractional alopecia, or anagen effluvium, as well as those with a diagnosis or history of oral ulcers, were excluded from the HC group.

The inclusion and exclusion criteria are shown in Figure 1. We set a high threshold for the recruitment criteria specified in Figure 1 to best standardize the confounders.

The common exclusion criteria for all groups were as follows: patients aged <18 or >50 years; those with current and/or previous history of deep vein thrombosis (DVT) and/or venous thrombosis other than DVT; those with chronic venous insufficiency (notably, 27 patients with venous insufficiency findings during CFV measurements with Doppler US were excluded); pregnant or breastfeeding individuals; those with systemic diseases such as obesity (body mass index [BMI] > 30 kg/m^2^); those with heart and lung diseases, hypertension, diabetes mellitus, cerebrovascular diseases, inflammatory bowel diseases, psychiatric diseases, or malignancy; those with hematological, rheumatological (except mucocutaneous BD), autoimmune, or autoinflammatory diseases; those with immunodeficiency; those with active infection; those with any organ involvement, including a history of thrombophlebitis secondary to BD; and those using additional drugs other than colchicine. Furthermore, a difference of >0.1 mm between the right and left CFV whole-wall and intima–media thicknesses was used as an exclusion criterion because of the possibility of a lateralized cardiovascular disease, which was not excluded. Accordingly, four participants were excluded from the study. The patient inclusion and exclusion flowchart in the study was presented in more detail in Figure 2.

### 2.3. Venous Doppler Ultrasonography

All Doppler US examinations were performed on the same day as the clinical assessment by a single expert radiologist with eight years of US experience who was blinded to the cases (Y.S.). Bilateral lower extremity veins (the CFV, superficial femoral vein, popliteal vein, and great saphenous vein) were examined in the craniocaudal direction using a high-resolution Doppler US system (Aplio i800; Canon Medical Systems, Tokyo, Japan) equipped with a high-resolution linear transducer (14–18 MHz). The hip was externally rotated, and the knee was slightly rotated to reveal deep veins and reduce muscle tension. Deep and superficial lower extremity veins were examined for the presence or absence of chronic thrombotic changes, venous insufficiency, recanalization, reflux, and collateral development. In supine and prone positions, venous insufficiency was assessed following the Valsalva maneuver in the saphenofemoral junction and popliteal veins. VWT and IMT of the CFV measurements were performed 2 cm distal to the saphenofemoral junction in the supine position from the posterior wall of the vessel to avoid reverberation artifacts. Anterior wall examination was not performed, as it may cause reverberation artifacts that would result in an imperfect delineation. IMT measurements were performed from the blood–intima interfaces to the outer end of the hypoechoic line (media–adventitia junction). The adventitial layer was excluded from the IMT measurement. Two measurements were taken from each vessel, and the average value was noted. WWT and IMT measurements of CFV are shown in Figure 3.

WWT and IMT of the CFV were measured on the same day by two different radiologists to determine ‘inter-observer reliability’ in eight patients, each with BD and RAS. They were measured again a few days later by the same radiologist to determine ‘intra-observer reliability’. It was observed that both interobserver and intraobserver concordances were good for the mean of the right and left WWT of the CFV [intraclass coefficient (ICC): 0.881 (95% CI: 0.712, 0.953), *p* < 0.001; ICC: 0.913 (95% CI: 0.812, 0.988), *p* < 0.001; respectively]. When we converted the WWT of the CFV to categorical evaluation according to the cutoff value, the concordance was 100% (κ = 1). Also, it was observed that both inter- and intraobserver concordances were good for the mean of the right and left IMT of the CFV [ICC: 0.731 (95% CI: 0.672, 0.822), *p* < 0.001; ICC: 0.733 (95% CI: 0.688, 0.838), *p* < 0.001; respectively].

### 2.4. Statistical Analysis

All procedures were performed using the Statistical Package for Social Sciences software (SPSS Inc., Chicago, IL, USA, v21.0) and G*Power 3.1. Results were presented as mean ± standard deviation and number (percentage). Priori power analyses in the ANOVA test were performed for medium effect size (f = 0.25), two-sided α = 0.05, and power (1 − β) = 0.90 at a confidence level of 95%. Accordingly, the required sample size was determined as a minimum of 207 patients. Pearson’s chi-square test was used for categorical variables. The Kolmogorov–Smirnov test checked the normality distribution of scale variables. Since the data did not represent normal distribution, nonparametric tests were preferred in all analyses. Independent samples were analyzed with appropriate significance tests (e.g., the Mann–Whitney U test and Kruskal–Wallis H test). Receiver operating characteristics (ROC) analysis was performed to evaluate the best cutoff value to distinguish BD from RAS. In addition, the contribution of the WWT of the CFV to the differentiation of BD was presented by univariate logistic regression analysis. The Kappa test was used for concordance analysis. The ICC was used to determine both intra- and interobserver reliability. A two-sided *p*-value < 0.05 was accepted as statistically significant. Bonferroni correction was performed post hoc after the Kruskal–Wallis H test.

## 3. Results

A total of 239 participants were included in the present study. The HC, RAS, and BD groups were matched in terms of the mean age of the participants (34 ± 10, 36 ± 11, and 37 ± 9 years, respectively) and sex distribution (female-to-male ratio: 0.71, 0.85, and 0.68, respectively). Furthermore, the study groups were adjusted for possible confounders such as BMI, waist circumference, smoking, and alcohol use in line with the purpose of the study. We found that the duration of complaints was significantly longer, and current colchicine use was more common in the BD group than in the RAS group (*p* = 0.011 and *p* < 0.001, respectively). The mean C-reactive protein and erythrocyte sedimentation rate levels were similar in all three groups (Table 1).

Statistical differences in the WWT and IMT of the CFV in the differential diagnosis of BD are presented in Table 2. Although there were no differences between the groups in terms of the IMT, the VWT differed between the BD group and the HC and RAS groups (*p* < 0.001).

The WWT of the CFV was subjected to ROC analysis, followed by logistic regression analysis. Notably, the results of the ROC analysis of the WWT of the CFV, which may be useful in the differential diagnosis of BD and RAS, are presented in Figure 4 and Table 3, with the area under the ROC curve (AUROC) value, cutoff value, *p*-value, sensitivity, specificity, and negative and positive likelihood ratios being presented. Furthermore, as shown in Table 3, the WWT of the CFV was useful in predicting BD in patients presenting with complaints of ROUs (AUROC = 0.800; *p* < 0.001). Moreover, the WWT of the CFV had an advantage in terms of sensitivity and specificity at a cutoff value of 0.58 mm (WWT of CFV > 0.58 mm: sensitivity = 79.8%, and specificity = 64.7%).

The odds ratio (OR) and significance level based on univariate logistic regression analysis are shown in Table 4. In patients presenting with complaints of ROUs and a WWT of the CFV of >0.58 mm, the probability of BD diagnosis rather than RAS diagnosis was significantly increased (OR = 6.418, 95% confidence interval: [3.44–11.96], *p* < 0.001).

## 4. Discussion

The present study reported that patients with BD had significantly higher WWTs of the CFV than healthy controls and patients with RAS. In this study, the IMT of the CFV was similar in all three groups. Moreover, the WWT and IMT of the CFV were similar between the RAS and HC groups. To the best of our knowledge, this is the first study to evaluate the use of the WWT and IMT of the CFV in patients with BD and RAS and to demonstrate the role of the WWT of the CFV in differentiating these two diseases.

Notably, BD is not a single unique entity, but a syndrome with different clinical phenotypes, and it can involve arterial and venous vessels of all sizes. Given that the clinical diagnostic criteria by the International Study Group (ISG) had low sensitivity, the International Criteria for Behçet’s Disease (ICBD) was developed in 2014 [21]. The ICBD criteria appear to be more sensitive, especially in early stages of the disease; however, they can lead to overdiagnosis, and in particular, patients with spondyloarthropathy features may be misdiagnosed with BD [21]. Recently, cases of incomplete BD have been increasing in Far Eastern countries, such as Japan and Korea [22]. Notably, in countries where the prevalence of BD is high, the diagnosis of BD is usually made by an expert, given the knowledge and experience of physicians regarding patients presenting with limited findings, such as oral aphthous lesions and major organ involvement, which do not meet the ISG criteria [14]. Conversely, in countries with a low prevalence of BD, its diagnosis can be challenging. The diagnosis of BD is primarily based on clinical symptoms after the exclusion of other potential causes, and there is no specific laboratory, histopathological, or genetic test for its diagnosis [23]. Although the sensitivity of the skin pathergy test is reported to be 40–80% in some regions, its accuracy decreases to 20% in some countries, thus reducing the diagnostic value of the test [24]. Furthermore, if BD is not active, the sensitivity of this test is much lower [25]. In the present study, the results of the pathergy test were positive for 35% of patients with BD. The diagnosis of BD is relatively straightforward when patients have clinical findings such as genital ulcers, ocular involvement, or pulmonary artery aneurysm, in addition to recurrent oral ulcers. Except for recurrent oral ulcers, BD is challenging to diagnose in patients with unclear clinical findings and is often confused with RAS. Based on this challenging situation in clinical practice, we designed our study as are shown in Figure 1 and Figure 2. However, patients with presumed RAS may be diagnosed with BD after several years. A study from Turkey reported that the time between developing an oral ulcer and meeting the diagnostic criteria for BD was approximately four years [26]. Thus, new methods are required to predict the conversion to BD over time in patients with RAS who do not fully meet the criteria for BD [7].

In BD, venous vessels are involved in 67–84% of all vascular manifestations, whereas arterial vessels are involved in less than 15% of cases. Superficial and deep vein thrombosis in the lower extremities is the most common form of vascular BD, which is seen in approximately 80% of the cases [9]. It can also occur in unusual sites such as the hepatic veins (Budd–Chiari syndrome), the inferior and superior vena cava, and the cerebral venous sinus. The involvement of the pulmonary artery, which is more like a vein than an artery, is also characteristic of BD [10]. The most important histopathological finding of BD is vasculitis with prominent neutrophil and monocyte infiltration in the perivascular region of the vessel wall, which may lead to the thickening of the vessel walls [27]. In BD, vein wall inflammation has been reported in cases of skin, ocular, and neurological involvement, in addition to cases of vascular involvement [9]. Ambrose et al. were the first to directly evaluate the veins in BD patients, and they reported an increased VWT in the popliteal veins of patients with BD, which were measured by magnetic resonance imaging [28]. Subsequently, Boulon et al. reported increased VWT in the right great saphenous vein, which was measured by US in a patient presenting with acute calf pain without thrombosis [29].

Notably, Alibaz-Oner et al. were the first to investigate the VWT of bilateral the CFV and great/small saphenous veins in patients with BD using Doppler US, and they reported that the VWT was significantly increased in patients with BD compared to patients with ankylosing spondylitis (AS) and healthy controls, regardless of the vascular involvement [12]. Furthermore, bilateral VWT measurements of the CFV have demonstrated approximately 80% sensitivity and specificity for cutoff values of 0.5 mm [12]. In another study, Alibaz-Oner et al. reported that the diagnostic performance of the VWT measurement of the CFV (cutoff = 0.5 mm) in the BD group compared with healthy controls with multiple diseases (such as AS, systemic vasculitis, antiphospholipid antibody syndrome, venous insufficiency, and noninflammatory DVT) was very good, with a sensitivity and specificity of >80% (except for antiphospholipid syndrome), regardless of the presence of any organ involvement [15]. Moreover, the VWT was not associated with acute phase response, disease activity, age, or disease duration [15]. Seyahi et al. reported significantly higher VWT measurements in patients with BD than in healthy controls [13]. In another study by Alibaz-Oner et al., diagnostic cutoff values of ≥0.5 mm for the VWT of the CFV showed good performance in differentiating patients with BD from patients with Crohn’s disease and healthy controls [14]. Similar to other studies in which the VWT was measured in BD (Appendix A), we found that the VWT of the CFV was significantly higher in the BD group than in the other study groups. While the WWT of the mean right CFV of BD patients in the present study was 0.68 ± 0.12 mm, it was found to be between 0.68–0.92 mm in other studies [11,12,13,14,15,16,17,18]. In the present study, the WWT of the CFV had an advantage in 79.8% sensitivity and 64.7% specificity at a cutoff value of 0.58 mm. Furthermore, the results of the present study indicate that the measurement of the WWT of the CFV using Doppler US (a noninvasive and easily accessible tool) may be a useful diagnostic tool in differentiating BD from RAS.

As indicated by the results of the above-mentioned studies, a significant increase in the VWT, even in cases of BD without venous thrombosis, suggests the presence of subclinical inflammation that is localized to venous structures. For example, carotid intima–media thickness (CIMT) is one of the most widely accepted biomarkers of subclinical atherosclerosis. Notably, in a previous study, there was no significant increase in the CIMT and femoral artery IMT in cases of BD [30]. Inflammation, which is thought to be the cause of increased wall thickness in vessels, is more prominent in veins than in arteries in BD cases [17]. Moreover, in cases presenting BD compared with arterial involvement, venous involvement affects the intimal layer and may cause significant changes in the vessel wall [17,31]. However, information regarding the histopathology of the venous involvement in BD is limited. In an autopsy series, three patients with BD and venous involvement exhibited intimal fibrous thickening and mild cellular infiltration in the vessel wall. In another autopsy study, one patient with BD and pulmonary thrombosis presented with thrombus covering a thickened fibrous intima and inflammatory cell infiltration mainly between the intima and thrombus [31]. In cases of BD, although there is no histological evidence, it has been reported that venous inflammation, which may cause endothelial dysfunction and a thrombosis tendency, may increase the venous wall thickness [10]. However, it is currently unknown whether venous inflammation is mainly mediated by vasa vasorum affecting the adventitia, as in arterial disease, or by endothelial inflammation causing intimal thickening [17]. In the relevant literature, the number of studies evaluating the venous IMT is limited. In the study by Atalay et al., which evaluated the VWT by measuring the IMT of various veins, including the CFV, in pediatric patients, reported that the median VWT of both patients with definite and incomplete BD was significantly higher in all bilateral veins compared with that in the control group [18]. Furthermore, Sevik et al. compared the WWT and IMT of the CFV using Doppler US in BD and HC groups. They found that the WWT and IMT of the CFV were increased in the BD group compared with the HC group [17]. However, in the present study, we found no significant difference between these groups regarding the IMT of the CFV. This contradiction may be attributed to various reasons. For example, a previous study reported IMT increases in patients exhibiting signs of venous insufficiency [32,33]. However, in the study by Sevik et al., venous insufficiency findings were not considered as an exclusion criterion [17]. Moreover, in the study of Bissacco et al., great saphenous vein (GSV) and CFV IMTs were increased in patients with GSV reflux compared to those without GSV reflux [32]. In another study, compared with healthy controls, the venous insufficiency was found to be higher in patients with BD without vascular involvement [34]. A possible reason for this contradiction is that we excluded 27 patients with venous insufficiency findings detected at the time of CFV measurements with Doppler US.

Highly different and wide ranges of venous IMT values (0.30–0.77 mm) have been reported in patients with and without BD [17,18,32]. This may be attributed to differences in the techniques and positioning adopted by the operator, the experience of the radiologist, and the study design. Moreover, there is confusion regarding the terminologies used when performing venous measurements with US. Although the VWT was evaluated by measuring the WWT in most studies in which venous measurements were performed with a US to detect BD, the VWT was evaluated by measuring the IMT in the study of Atalay et al. Thus, an accurate and acceptable radiological technique must be developed to minimize measurement-related limitations and subjectivity. Studies comparing different disease groups with BD, in which the IMT and whole VWT measurements are evaluated together, are needed.

The main limitations of the present study include its cross-sectional design and use of a US, which is an operator-dependent modality. Our study had a relatively small sample size. However, the number of patients was comparable to that of other studies with vein wall thickness measurements, and we had more patients than in some other studies (Appendix A). Our primary objective was to compare mucocutaneous BD and RAS patients with recurrent oral ulcers, so BD patients with organ involvement were not included in our study. Measurements were made by the same radiologist who did not know the patient’s diagnoses, and our intraobserver and interobserver reliability data could be evaluated with a small number of patients. Studies with a longitudinal design and a long follow-up period might reveal whether our observations persist throughout the course of the disease.

## 5. Conclusions

In conclusion, the present study is noteworthy, as it reported that US may be useful as a novel tool for measuring the VWT of the CFV to differentiate RAS and BD, which is often quite challenging in clinical practice. Furthermore, it reported that the WWT of the CFV (>0.58 mm) may be useful in distinguishing BD from RAS, with 79.8% sensitivity and 64.7% specificity; however, these results need to be supported by further comprehensive studies.

## Figures and Tables

**Figure 1 diagnostics-13-02705-f001:**
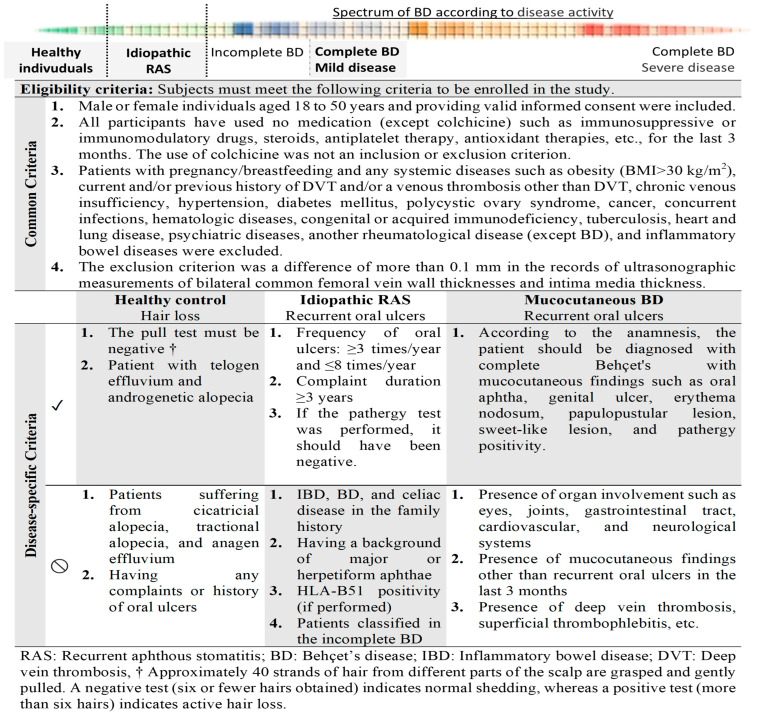
Inclusion and exclusion criteria of the study.

**Figure 2 diagnostics-13-02705-f002:**
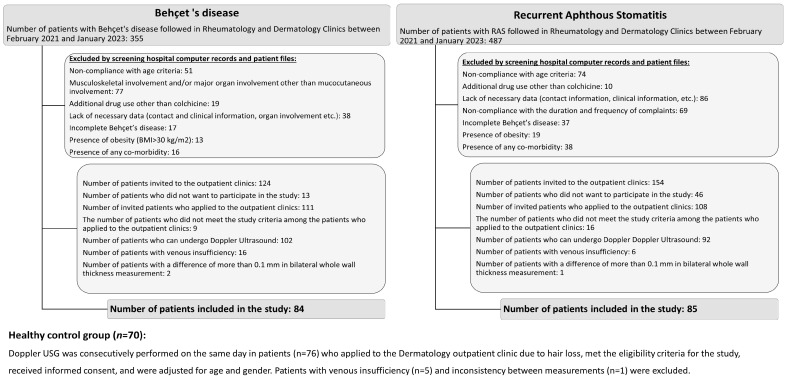
The patient inclusion and exclusion flowchart.

**Figure 3 diagnostics-13-02705-f003:**
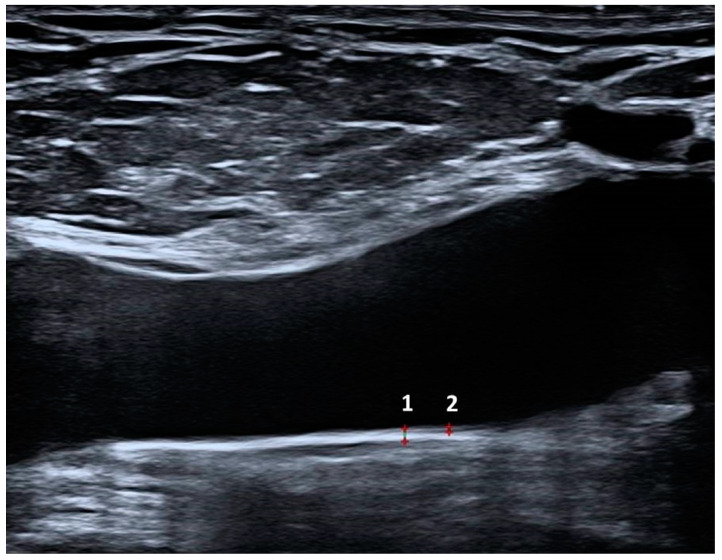
The measurement of the common femoral vein whole-wall (1) and intima–media thickness (2) in a patient with Behçet’s Disease.

**Figure 4 diagnostics-13-02705-f004:**
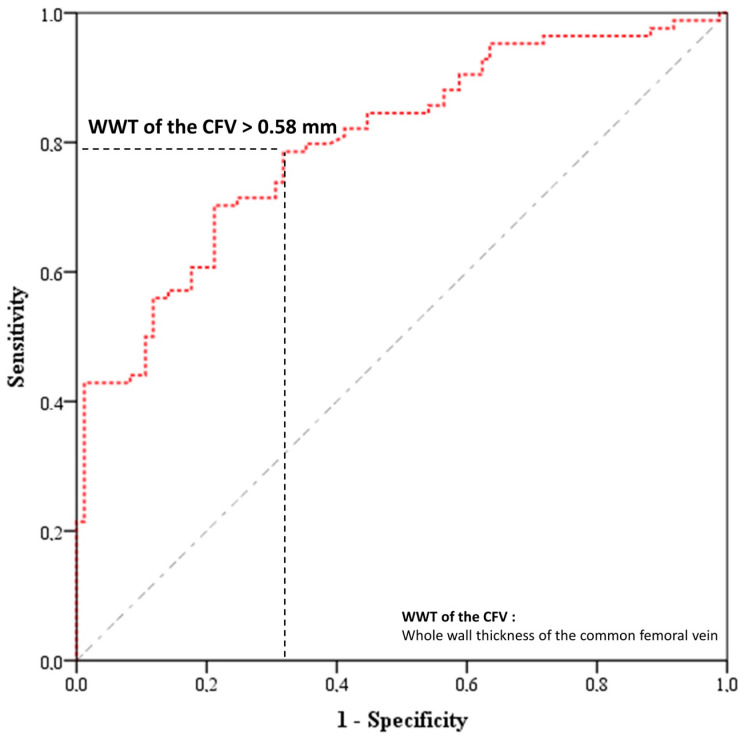
Evaluation of the WWT of CFV measurement with receiver operating characteristic curve graph in the distinction of RAS and BD.

**Table 1 diagnostics-13-02705-t001:** Evaluation of potential confounders between groups in the study.

Confounders		Healthy Control	Idiopathic RAS	Behçet’s Disease	*p*-Value
		(*n* = 70)	(*n* = 85)	(*n* = 84)	
Age (years)		34 ± 10	36 ± 11	37 ± 9	0.065
Sex	Man	41 (58.6%)	46 (54.1%)	50 (59.5%)	0.753
	Woman	29 (41.4%)	39 (45.9%)	34 (40.5%)	
BMI (kg/m^2^)		25.8 ± 3.0	24.9 ± 3.2	25.1 ± 3.2	0.059
Waist circumference (cm)		92 ± 8	89 ± 16	91 ± 12	0.069
The duration of complaints (year)		-	4.5 ± 1.7	4.7 ± 1.8	0.120
Colchicine usage, currently	Yes	-	23 (27.1%)	70 (83.3%)	**<0.001 ***
	No		62 (72.9%)	14 (16.7%)	
Colchicine duration (year), if yes		-	1.9 ± 1.3	2.6 ± 3.7	0.377 *
Smoking	Yes, currently	8 (11.4%)	9 (10.6%)	9 (10.7%)	0.621
	Ex smoker	8 (11.4%)	4 (4.7%)	6 (7.1%)	
	No, never	54 (77.1%)	72 (84.7%)	69 (82.1%)	
Total cigarette exposure (pack/year)		16.3 (10.5)	8.2 (9.4%)	13.5 (10.7)	0.104
Alcoholic beverage consumption	Yes	9 (12.9%)	8 (9.4%)	7 (8.3%)	0.360
	No	61 (87.1%)	77 (90.6%)	77 (91.7%)	
**Others**					
Pathergy test	Not done	70 (100%)	-	-	N/A
	Negative	-	85 (100%)	54 (64.2%)	
	Positive	-	0 (0%)	30 (35.7%)	
ESR (mm/hour)		10 ± 13	18 ± 15	16 ± 14	0.061
CRP (mg/l)		2.6 ± 2.5	10.9 ± 19.9	6.7 ± 11.8	0.296

RAS: Recurrent aphthous stomatitis; BMI: Body mass index; N/A: Not applicable; ESR: Erythrocyte sedimentation rate; CRP: C-reactive protein. Data are expressed as mean ± standard deviation—Kruskal–Wallis H, Mann–Whitney U (*), and Pearson’s chi-square tests were used. Significant values were shown in bold.

**Table 2 diagnostics-13-02705-t002:** Evaluation of statistical differences in the WWT and IMT of CFV in the differential diagnosis of BD.

Ultrasonographic Parameters		Healthy Control	Idiopathic RAS	Behçet’s Disease	*p*-Value
		(*n* = 70)	(*n* = 85)	(*n* = 84)	
**WWT of CFV** (mm)					
Right		0.56 ± 0.08	0.54 ± 0.09	0.68 ± 0.12	**<0.001** ^a,b^
Left		0.57 ± 0.10	0.55 ± 0.10	0.68 ± 0.11	**<0.001** ^a,b^
	Mean of right and left	0.56 ± 0.09	0.54 ± 0.09	0.68 ± 0.11	**<0.001** ^a,b^
**IMT of CFV** (mm)					
Right		0.245 ± 0.018	0.234 ± 0.030	0.242 ± 0.044	0.096
Left		0.249 ± 0.026	0.239 ± 0.033	0.241 ± 0.041	0.211
	Mean of right and left	0.247 ± 0.019	0.237 ± 0.030	0.241 ± 0.041	0.116

RAS: Recurrent aphthous stomatitis; CFV: Common femoral vein; IMT: Intima–media thickness; WWT: Whole-wall thickness. Data are expressed as the mean ± standard deviation. Kruskal–Wallis H and Mann–Whitney U (post hoc) tests were used. Bonferroni correction was applied post hoc after the Kruskal–Wallis H test. Significant values are shown in bold. Note: There was no difference between the healthy control group and the RAS group (*p* > 0.05). a: *p* < 0.05 for the difference between the control group and Behçet’s disease group. b: *p* < 0.05 for the difference between the RAS group and the Behçet’s disease group.

**Table 3 diagnostics-13-02705-t003:** Receiver operating characteristic curve analysis of the WWT of CFV to distinguish BD from RAS in patients with frequent oral aphthae.

	AUROC	Cutoff Value	Sensitivity	Specificity	Pos LR	Neg LR	Accuracy Rate	*p*-Value
**WWT of CFV** (mm)	0.800	>0.58	79.8%	64.7%	2.26	0.31	72.2%	<0.001

WWT of CFV: Whole-wall thickness of the common femoral vein based on the mean of the right and left femoral vein measurements. AUROC: Area under the receiver operating characteristic curve; Pos LR: Positive likelihood ratio; Neg LR: Negative likelihood ratio; Significant value is shown in bold.

**Table 4 diagnostics-13-02705-t004:** Evaluation of WWT of CFV related to BD by univariate binary logistic regression analysis.

	βi	Odds Ratio	95% CI		Wald Value	Cox and Snell R^2^	Nagelkerke R^2^	*p*-Value
			Lower	Upper				
**WWT of CFV > 0.58 mm**	1.86	6.41	3.44	11.96	34.15	0.154	0.212	**<0.001**

Βi: Regression coefficient; CI: Confidence interval; WWT of CFV: Whole-wall thickness of common femoral vein based on the mean of the right and left femoral vein measurements.

## Data Availability

Detailed data are available if requested from the corresponding author.

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
