# Peer review of "Measuring the Whole Wall Thickness of the Common Femoral Vein as a Distinctive Diagnostic Tool to Distinguish Behçet’s Disease Presenting with Oral Ulcers from Recurrent Aphthous Stomatitis"

_diagnostics, 2023, doi:10.3390/diagnostics13162705_

Round 1

Reviewer 1 Report

This paper is well written, but there are some unclear points, so please let me comment or ask questions as follows.

1. Regarding the revision of the Declaration of Helsinki, the latest version is 2013, not 2008. Please check this.

2. Please add how many patients visited the hospital during this study period, and how many of them consented to participate in this study.

3. Please state how many BD or RAS patients met the inclusion and exclusion criteria in this study.

4. The number of patients in the three groups was similar (n = 70, 85, or 84). How was the number of patients for this study determined?

5. Regarding 2-4 above, it might be good to illustrate this as a flowchart.

6. Am I correct in understanding that a history of medications such as glucocorticoids, immunosuppressants, or biologics is not an exclusion criteria?

7. Age, BMI and waist circumference are likely to show significant differences between the three groups as the number of subjects increases. In terms of study design, it would be desirable for these items not to be significantly different between the three groups, but is this a limitation of the study?

8. Please add glucocorticoids, immunosuppressants, and biologics to Table 1.

9. Tables 3 and 4 are the same, so please review them.

10.  Please explain the figures and tables in this study in the discussion

Author Response

Dear Reviewer

We would like to thank you for your valuable comments and suggestions for the early version of our manuscript. We are grateful for your time and consideration. We have revised the manuscript in accordance with the specific and general requests. Responses are attached below.

 Best regards,

Reviewer 2 Report

This study assesses the utility of ultrasonographic technology to measure vascular thickness as an indicator of behcets clinical activity. The sensitivity and specificity of the procedure as measured in the study are moderate and suggestive that further studies are warranted with a larger n and perhaps with technological improvements in the future. The study is interesting but the technology as currently described is still far from ready to be implemented for clinical practice. 

Major quality improvements are needed as far as English language usage throughout the manuscript. 

Author Response

Dear Reviewer

We would like to thank you for your useful comments and suggestion given for the early version of our manuscript. We are grateful for your time and consideration.

English language editing has been done according to your request. I emailed the English editing certificate to the Diagnostics Editorial Office. If necessary, we can have English language editing done by MDPI English Editing Services.

We agree with you that the sensitivity and specificity of the results of our study were moderate and that more comprehensive studies are needed on this subject; we stated this in conclusion part of our study (Furthermore, it reported that WWT of CFV (>0.58 mm) may be useful in distinguishing BD from RAS, with 79.8% sensitivity and 64.7% specificity; however, these results need to be supported by further comprehensive studies).

Behcet Syndrome Activity Score (BSAS) was not evaluated because our study aimed to differentiate between RAS and BD disease presenting with oral ulcers by measuring the wall thickness of the common femoral vein with Doppler ultrasound. However, in patients with Behcet's disease (BD), no correlation was observed in the previous studies between bilateral CFV thickness and age, disease duration, BMI, acute phase reactants, or the Behcet Syndrome Activity Score (BSAS) [15]. Many studies have shown increased venous wall thickness in BD patients independent of vascular involvement [13,17,18]. Therefore, in most rheumatology clinics in our country, femoral vein wall thickness measurement is performed with Doppler ultrasound in the diagnosis and differential diagnosis of BD disease. In clinical practice, dermatologists encounter RAS patients much more frequently than rheumatologists. With the results of our study, we think that venous wall thickness measurement with Doppler ultrasound can be used more frequently in the differentiation of RAS-BD in dermatology clinical practice. Moreover, Doppler USG is an easily available and non-invasive test and can be used as an auxiliary diagnostic tool, especially in patients with suspected Behçet's disease.

Best regards,

Round 2

Reviewer 1 Report

Thank you for the good revisions of your manuscript. This paper deserves to be published. I wish you good luck.

Reviewer 2 Report

All of critiques were adequately addressed in the manuscript. There are no other concerns noted. I recommend to accept manuscript as is.